# COVID-19 Therapeutics Use by Social Deprivation Index in England, July 2020–April 2023

Angela Falola [1,*], Hanna Squire [1], Sabine Bou-Antoun [1], Alessandra Løchen [2], Colin S. Brown [1,3] and Alicia Demirjian [1,4]

1    Healthcare-Associated Infection (HCAI), Fungal, Antimicrobial Resistance (AMR), Antimicrobial Use (AMU) & Sepsis Division, United Kingdom Health Security Agency (UKHSA), London NW9 5EQ, UK
2    Tuberculosis, Acute Respiratory Infections, Zoonosis, Emerging Infections and Travel Health Division, United Kingdom Health Security Agency (UKHSA), London NW9 5EQ, UK
3    NIHR Health Protection Research Unit in Healthcare Associated Infections and Antimicrobial Resistance, Imperial College London, London SW7 2AZ, UK
4    Department of Paediatric Infectious Diseases & Immunology, Evelina London Children's Hospital, UK Faculty of Life Sciences & Medicine, King's College London, London SE1 7EH, UK
*    Correspondence: angela.falola@ukhsa.gov.uk

**Abstract:** Coronavirus disease-19 (COVID-19) has disproportionately affected certain demographics in England, exacerbating existing health disparities. Effective therapeutics are a critical line of defence against COVID-19, particularly for patients at elevated risk for severe disease. Surveillance systems were established to monitor the usage of COVID-19 therapeutics in hospital and community settings and to inform stewardship. Three antiviral therapies—nirmatrelvir plus ritonavir (Paxlovid®), remdesivir (Veklury®), and molnupiravir (Lagevrio®)—and two neutralising monoclonal antibody therapies (nMAbs)—sotrovimab (Xevudy®) and casirivimab with imdevimab (Ronapreve®)—were in use in England between July 2020 and April 2023. This paper aims to illuminate trends in the utilisation of COVID-19 therapeutics treatment in both hospital and community settings, stratified by the Index of Multiple Deprivation (IMD) in England. Chapter 3 of the English Surveillance Programme for Antimicrobial Utilisation and Resistance (ESPAUR) Report 2022 to 2023 also discusses the epidemiological surveillance of these five directly acting antiviral COVID-19 therapeutics' use in England between 2022 and 2023.

**Keywords:** antivirals; community; COVID-19; deprivation; hospital; therapeutics

## 1. Introduction

Effective therapeutics have become a critical line of defence against coronavirus disease 2019 (COVID-19), caused by the severe acute respiratory syndrome coronavirus 2 (SARS-CoV-2), particularly for patients at elevated risk for severe disease [1,2]. To accelerate the discovery and distribution of antiviral therapeutics, the UK government established the Antivirals Taskforce (ATF) [3]. Shortly thereafter, new and repurposed therapeutic agents for treating COVID-19 were introduced in April 2020. The United Kingdom first administered treatments in hospital settings on 11 July, 2020 [4] and initiated non-hospital (community) treatments on 16 December of the same year [3–5]. COVID-19 Medicine Delivery Units (CMDUs) were launched to facilitate the provision of COVID-19 therapeutics to non-hospitalised, high-risk patients in England [6,7]. Three antiviral therapies—nirmatrelvir plus ritonavir (Paxlovid®), remdesivir (Veklury®), and molnupiravir (Lagevrio®)—and two neutralising monoclonal antibody therapies (nMAbs)—sotrovimab (Xevudy®) and casirivimab with imdevimab (Ronapreve®)—were deployed to treat COVID-19 cases (Supplementary Materials). Corticosteroids and other immunomodulators were considered outside of the scope of this analysis. Surveillance systems were set up to monitor treatments in both hospital and community settings and assist with stewardship of these

therapeutics [8,9]. As in the previous year, the latest English Surveillance Programme for Antimicrobial Utilisation and Resistance (ESPAUR) 2022 to 2023 indicates variations in therapeutic requests between hospital and non-hospital settings [10], while casirivimab with imdevimab (available in the UK from September 2021 to February 2022) and remdesivir (available from May 2020) were predominantly used in hospitals for people with severe COVID-19 (Supplementary Materials).

Sotrovimab (launched in the UK mid-December 2021), nirmatrelvir plus ritonavir (launched in February 2022), and molnupiravir (introduced between mid-December 2021 and February 2022) were mainly administered in community settings to people with COVID-19 who had a high risk of severe outcomes to prevent their symptoms from worsening [6,9]. Though sotrovimab can be used in both community and hospital settings, it is mainly administered in the community setting.

COVID-19 has disproportionately affected certain demographics in England, exacerbating existing health disparities. Mortality rates in the most deprived areas surpass those in less deprived areas [6,7,11]. There have been efforts to address health inequalities in England; Core20PLUS5 is an NHS England initiative aimed at addressing and narrowing these health disparities [12]. Identifying a core target population, the approach focusses on five clinical areas requiring urgent improvement. Such strategies aimed at addressing health inequalities are critically important and necessitate ongoing stewardship. This paper aims to illuminate trends in the utilisation of COVID-19 therapeutics treatment in both hospital and community settings, stratified by the IMD in England [13].

## 2. Methods

### 2.1. Data and Data Sources

COVID-19 therapeutics usage data were sourced from NHS England via the Blueteq system, which collects patient-level treatment request data. The Blueteq system manages high-cost drugs for NHS England and, as such, contains clinical requests made for neutralising monoclonal antibodies (nMAB) and antiviral therapies used for the treatment of patients with COVID-19. NHS England interrogates whether clinical criteria are met before prescribing of antivirals can be undertaken. The data was grouped by month for the period between 11 July 2020 and 30 April 2023 in England. Requests for patient neutralizing monoclonal antibodies (nMAB) and antiviral therapies of interest are recorded in the Blueteq system, with data extracts received by UKHSA up to 30 April 2023. The treatment request data were cleaned (duplicate entries and invalid postcodes were excluded). As data on the number of patients eligible to receive COVID-19 therapeutics were unavailable to UKHSA, COVID-19 case counts were used as the denominator and were sourced from the UKHSA dashboard [14]. Cases with a valid home postcode were categorised into deciles according to the IMD 2019 in England [13], with the most deprived areas categorised as the 1st decile and the least deprived, the 10th decile. Deciles are calculated by ranking the 32,844 neighbourhoods in England and dividing them into 10 equal groups from most to least deprived. Within the data provided by the Blueteq system, there are three main categories for the setting variable, which are hospitalisation, hospital-onset, and community. The hospitalisation setting described cases hospitalised specifically for the management of COVID-19. Cases of COVID-19 initially identified in patients during their hospital stay were categorized as hospital-onset, and the community setting referred to non-hospitalised cases of COVID-19 in England.

### 2.2. Data Processing and Statistical Analyses

The first outcome was calculated using the total number of each therapeutic treatment request by quarter and year. Rates of treatment requests were measured as the number of requests per 100,000 COVID-19 cases by therapeutic and IMD decile. To estimate rates of total treatment requests by IMD between July 2020 to April 2023, the number of treatment requests were divided by the number of COVID-19 cases, normalised to 100,000 individuals by IMD decile. Rates of treatment requests were also grouped by setting into hospital-onset,

hospitalisation, and community; the number of treatment requests was divided by the number of COVID-19 cases, normalised to 100,000 individuals by setting and IMD decile.

All analyses were conducted using STATA 17 and R.

## 3. Results

In England, between July 2020 and April 2023, a total of 181,674 requests for therapeutic treatments comprising nMAB and antivirals against COVID-19 was recorded on the Blueteq system. More than half of these requests (56.7%, 103,048 treatments) were made within the community setting. Hospital settings accounted for the remaining 43.3% of treatment requests, with 35.6% (64,598 treatments) requested for those patients hospitalised for the management of COVID-19 and 7.7% (14,028 treatments) for those with hospital-onset COVID-19.

The highest numbers of treatment requests were in Quarter 1 and Quarter 2 2022 (n = 35,596; n = 26,332 treatments respectively). In 2022, there was a total of 108,786 treatment requests. Remdesivir was the most-prescribed treatment, constituting 36.8% (n = 66,946) of all requests. This was followed by nirmatrelvir plus ritonavir at 24.8% (n = 45,107), sotrovimab at 23.1% (n = 41,911), molnupiravir at 12.4% (n = 22,489), and casirivimab with imdevimab at 2.9% (n = 5221), the last of which was only used in England between Q3 2021 and Q1 2022 (Figure 1). Peaks in usage were seen in Q4 2020 for remdesivir (n = 16,484), in Q4 2021 for casirivimab with imdevimab (n = 4774), in both Q2 and Q3 2022 for nirmatrelvir plus ritonavir (n = 10,107 and n = 10,368, respectively), and in Q1 2022 for sotrovimab (n = 17,666), as indicated in Figure 1.

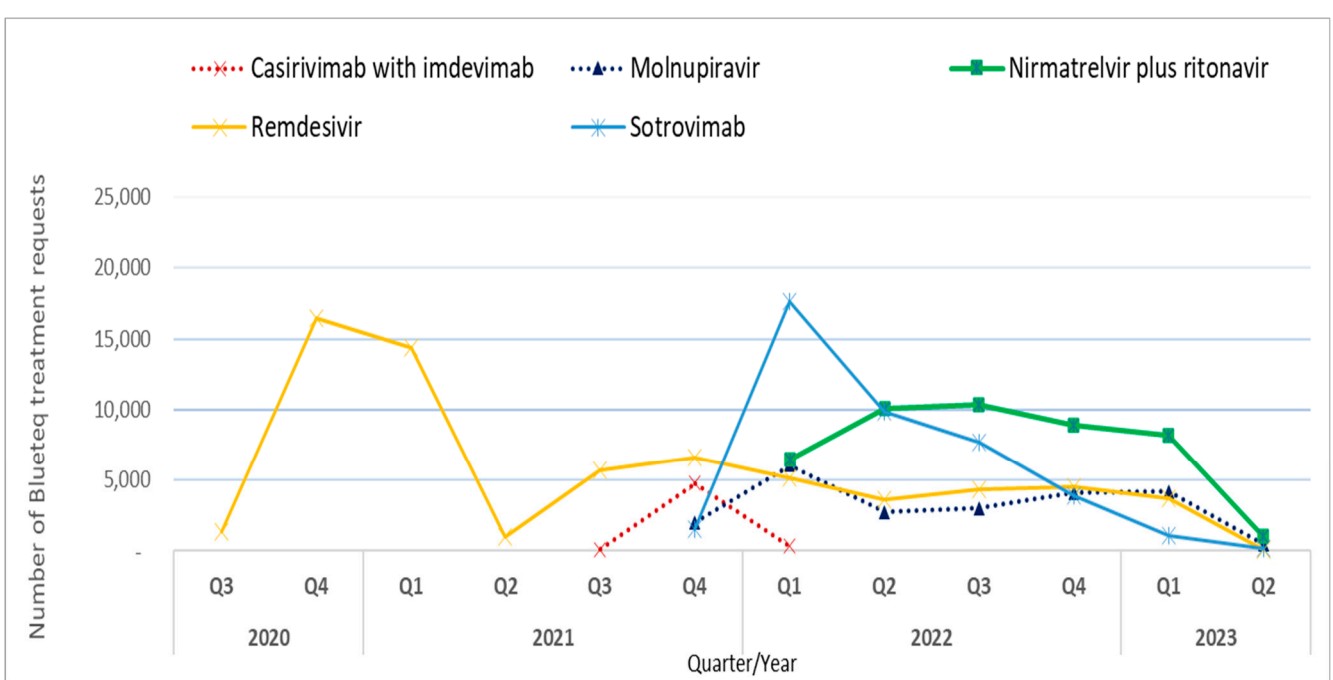

**Figure 1.** Number of treatment requests in Blueteq by therapeutic between Quarter 3 2020 and Quarter 2 2023. Note numbers < 11 are masked.

The rate of therapeutic use differed substantially according to IMD scores in England, and differentially according to the site of administration (Figure 2). The treatments primarily used in community settings—sotrovimab, molnupiravir, and nirmatrelvir plus ritonavir—displayed substantially lower usage rates in areas with higher deprivation compared to less deprived deciles. Conversely, remdesivir and casirivimab with imdevimab, predominantly used in hospital settings, demonstrated the opposite trend, and were requested at higher rates in more deprived areas. To illustrate, casirivimab with imdevimab had a higher rate of use in the first decile than in the least deprived decile areas (67.9;

39.9 per 100,000 COVID-19 cases, respectively). Similarly, remdesivir saw a higher usage rate in the most deprived decile compared to the least deprived decile (2359.3; 1633.6 per 100,000 COVID-19 cases, respectively).

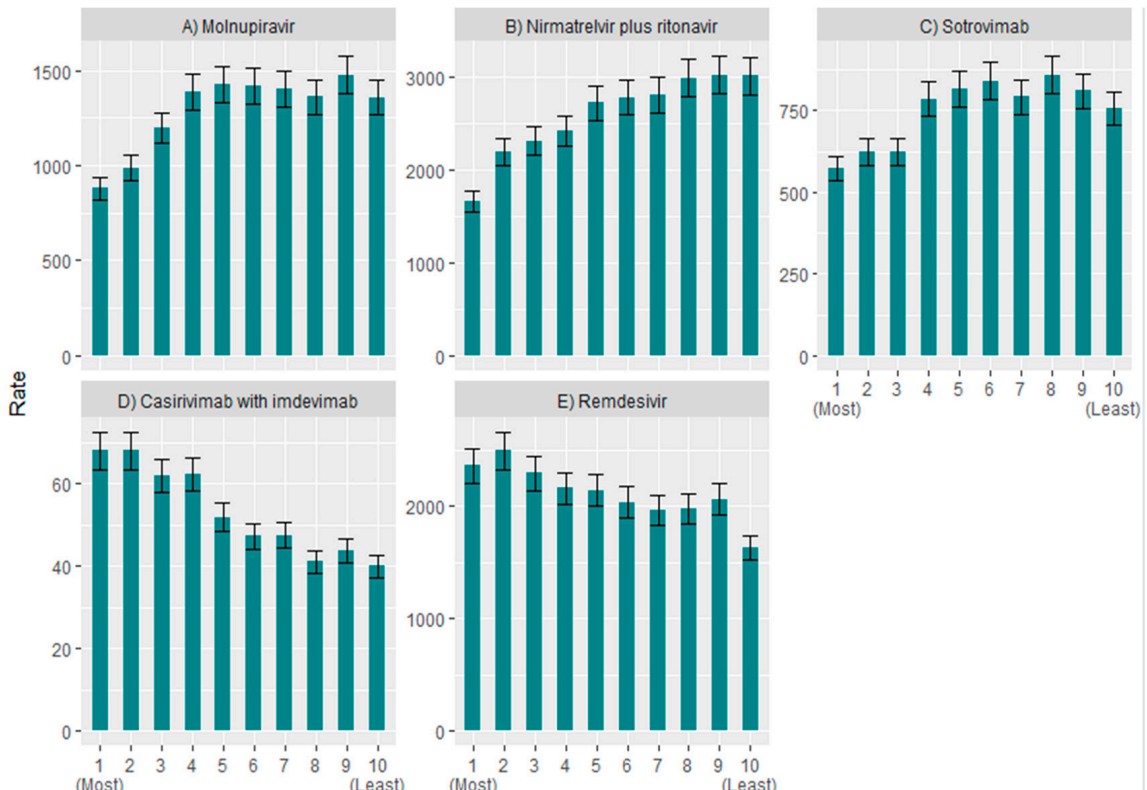

**Figure 2.** Rate of Blueteq treatment requests per 100,000 COVID-19 cases in England by therapeutic and IMD decile (most (1) to least (10) deprived). 1. Therapeutics in (**A**–**C**) are predominately used within the community setting. 2. Therapeutics in (**D**,**E**) are predominantly used within the hospital setting. Note differences in scale.

Between April 2020 and March 2023, therapeutic treatments that were predominantly administered in community settings constituted the largest proportion of treatments. Specifically, nirmatrelvir plus ritonavir (3015.8; 1666.1 per 100,000 COVID-19 cases, respectively) exhibited a higher rate of use in the least deprived decile areas compared to the most deprived decile. Molnupiravir (1355.5; 878.8 per 100,000 COVID-19 cases, respectively) and sotrovimab (756.9; 572.2 per 100,000 COVID-19 cases, respectively) followed suit, with higher rates in the least deprived decile when compared to the most deprived first decile. It should be noted, however, that these rates are subject to fluctuations over time.

As seen in Figure 3, the rate of therapeutic use varied by IMD decile across different settings. Of the total treatment requests per 100,000 COVID-19 cases, 42,716.3 were made within the community setting; for patients hospitalised for COVID-19, there were 17,364.0 requests; and for hospital-onset COVID-19, 7871.4 requests were made. Rates of therapeutic use within the community setting were much lower in the most deprived areas than in the least deprived areas, with 53.3% higher rates in the least deprived decile than in the most deprived decile (4852.4; 2585.1 per 100,00 COVID-19 cases, respectively). However, within the hospital setting, there was a reverse trend, which indicates higher usage in the least deprived areas than in the most deprived areas. This accounts for a 67.1% difference between the most deprived and the least deprived areas (950.3; 637.7 per 100,000 COVID cases, respectively), while the rate of requests for individuals hospitalised for COVID-19 within the most deprived decile was 65.3% higher than that for patients within the least deprived decile (2008.9; 1311.6 per 100,000 COVID cases).

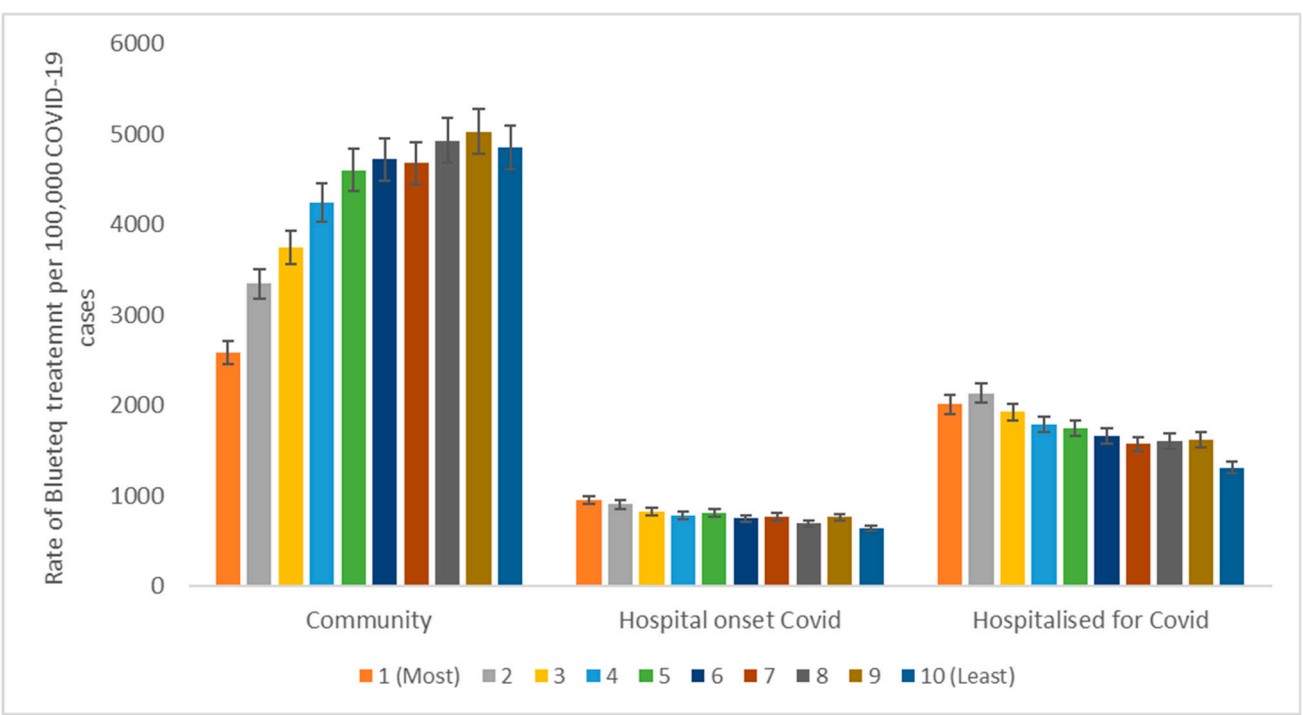

**Figure 3.** Rate of Blueteq treatment request per 100,000 COVID-19 cases in England by setting and IMD decile (most (1) to least (10) deprived).

## 4. Discussion

This study represents the first analysis to describe national trends in the use of therapeutic treatments for COVID-19 within both hospital and community settings, stratified by deprivation level in England. Treatment requests across both settings have exhibited fluctuations over time, especially in light of the introduction of new treatments, changes in viral strain, efficacy of therapy, and evolving guidelines.

A notable finding is the disparity in treatment rates between community and hospital settings across different deprivation deciles. Treatments in community settings were more prevalent among the least deprived deciles, while hospital settings saw a higher rate of treatments in the most deprived deciles over this period. It is important to caution that, without data on individual patients who were eligible for, offered, and completed each treatment, these trends may not necessarily reflect underlying inequities in access to healthcare.

This study is in line with another research effort that examined the breakdown of therapeutic treatment by key demographics and deprivation quintiles [5]. That study indicated that 12% of non-hospitalised patients receiving treatment resided in the most deprived areas, compared to 21% in the least deprived areas [5]. The higher rates of treatments in the most deprived deciles within the hospital setting could be attributed to several interconnected factors such as varying SARS-CoV-2 exposure risks, with lower-income populations often living in environments with higher exposure to pollutants and in overcrowded conditions, which may affect their risk of complications from respiratory infections. Differences in healthcare accessibility could lead to delayed diagnoses and treatments, which may contribute to these observed disparities, necessitating further research. Evidence from previous studies suggests some ethnic groups are more likely to live in more deprived areas than other counterparts [14]. Additionally, a higher level of deprivation is related to poorer health outcomes [15]. Delays in access to health care may result in delays in intervention, which could result in higher rates of hospitalisation. This could explain why remdesivir and casirivimab with imdevimab, which are predominantly used in hospital settings, were used more frequently among the most deprived deciles

compared to the least deprived, with the reverse pattern observed in community settings for molnupiravir, nirmatrelvir plus ritonavir, and sotrovimab rates.

Therapeutics primarily administered in community settings constituted most of all treatment requests between July 2020 and April 2023. Variations in treatment rates by IMD may also have geographical implications concerning where these drugs are most frequently prescribed. It is worth noting that not all these therapeutic agents were available throughout the entire period between July 2020 and April 2023. For instance, sotrovimab was incorporated into the clinical commissioning policy in mid- December 2021. Similarly, nirmatrelvir plus ritonavir was included in the clinical commissioning policy only as of February 2022. Additionally, free community testing was discontinued in April 2022.

The effectiveness of casirivimab with imdevimab against the dominant Omicron variant of SARS-CoV-2 has been uncertain, as most studies were carried out before this variant circulated. This resulted in limited requests for casirivimab with imdevimab during the winter of 2021–2022 and its eventual removal from the clinical commissioning policy in 2022 [8].

The limitations of this analysis include the absence of denominator data on individuals who were eligible for, offered, and accepted each treatment. Not all treatment requests may have resulted in patients receiving or completing treatment with these drugs, as the electronic or patient prescribing data were not accessible. As such, this study serves as an exploratory analysis and may not accurately depict variations or inequities in access to healthcare.

## 5. Conclusions

This paper highlights trends in COVID-19 therapeutics treatment by setting and deprivation decile. This study demonstrates differences in COVID-19 treatment-request rates across deprivation levels and settings. The most deprived groups accessed treatments that were predominantly dispensed in hospital settings at higher rates compared to the least deprived groups. This suggests that people living in more deprived areas experience more severe COVID-19 disease outcomes. Improving community access to health care within the most deprived areas would likely improve health outcomes and reduce the number of patients receiving treatments in hospital settings. Continued surveillance in monitoring new treatments, including information on the population eligible to receive these, is important to understand the nuances between settings and the potential patterns that are likely to emerge. Further research would be beneficial to examine the factors leading to high treatment rates in these two settings and to high hospital-admission rates generally.

**Supplementary Materials:** The following supporting information can be downloaded at: https://www.mdpi.com/article/10.3390/covid4050043/s1. Timeline for COVID-19 therapeutic Blueteq treatments.

**Author Contributions:** Conceptualisation, A.F.; methodology, A.F. and H.S.; validation, A.F., H.S., S.B.-A. and A.L.; formal analysis: A.F. and H.S.; investigation, A.F., H.S., S.B.-A. and A.D.; writing—original draft preparation, A.F. and H.S.; writing—review and editing. A.F., H.S., S.B.-A., A.D., A.L. and C.S.B.; supervision, S.B.-A., H.S. and A.D. All authors have read and agreed to the published version of the manuscript.

**Funding:** This research received no external funding.

**Institutional Review Board Statement:** This study was conducted as part of routine work; no ethical approval was required.

**Informed Consent Statement:** Not applicable.

**Data Availability Statement:** The data presented in this study are available in [Bou-Antoun, S.; Falola, A.; Budd, E., H.; Squire, H.; Brown, C.S.; Hope, R., Hopkins, S.; Muller-Pebody, B., Demirjian, A. The English Surveillance Programme for Antimicrobial Utilisation and Resistance (ESPAUR) report 2022 to 2023, Chapter 3 Antimicrobial Consumption, London, 2023].

**Acknowledgments:** Special thanks to NHS England colleagues for supplying the Blueteq data.

**Conflicts of Interest:** The authors declare no conflicts of interest.

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
