# Peer review of "COVID-19 Therapeutics Use by Social Deprivation Index in England, July 2020–April 2023"

_covid, doi:10.3390/covid4050043_

Round 1
Reviewer 1 Report
Comments and Suggestions for Authors
The manuscript provides an analysis of COVID-19 therapeutic utilization in England, emphasizing disparities in treatment access among different demographic groups. The authors effectively underscore the significance of surveillance systems in monitoring therapeutic usage and addressing health disparities exacerbated by the pandemic, by setting and deprivation.
Overall, the manuscript contributes to our understanding of COVID-19 therapeutic utilization and its implications for health disparities in England. The research is well-presented, and the message is effectively conveyed. I have no further comments for the authors.
Author Response
Dear Reviewer,
Thank you for reviewing our paper and for providing valuable feedback. Please find attached our revised manuscript.
Yours sincerely,
Angela

Reviewer 2 Report
Comments and Suggestions for Authors
Thank you for the opportunity to review this manuscript
some minor comments
Please align the manuscript to the RECORD-PE guidelines or other reporting guideline you consider appropriate
Please explain the IMD ranking system in the methods
Provide info on the setting to make it easier for the reader – some info on the health system is missing
Explain the coverage of database you used
Explain your main outcome measure, the variables used and the analysis in more detail
Make conclusion more specific to your findings
Author Response
Dear Reviewer,
We would like to thank you for the opportunity to submit a revised copy of the manuscript above (covid-2942098).
Following the reviews, we have carefully considered and addressed each comment. A summary of our response is outlined below, with corresponding changes highlighted in yellow in the revised manuscript.
Thank you again for your considering our revised manuscript.
Reviewer 2
|
Section |
Comment |
Response |
|
Methods |
1. Please align the manuscript to the RECORD-PE guidelines or other reporting guideline you consider appropriate. |
Where applicable sections in methods adjusted to RECORD-PE guidelines |
|
Methods |
2. Please explain the IMD ranking system in the methods |
Amended and explained in line 83-85 |
|
Methods |
3. Provide info on the setting to make it easier for the reader – some info on the health system is missing |
Added in line 85 to 90 |
|
Methods |
4. Explain the coverage of database you used |
Added in line 74 to 79 |
|
Methods |
5. Explain your main outcome measure, the variables used and the analysis in more detail |
Explained and added in line 92 to 93. Also, in line 97 to 99 |
|
Conclusion |
6.Make conclusion more specific to your findings |
Amended in line 217 to 218 |
Editorial Office
|
Section |
Comment |
Response |
|
Institutional Review Board Statement |
Please add |
Added in line 235 |
|
Informed Consent Statement |
Please add |
Added in line 237 |
Yours sincerely,
Angela Falola
